# Reputation Effect on Contract Choice and Self-Enforcement: A Case Study of Farmland Transfer in China

**Hanning Li** [1], **Hongyun Han** [2,*] **and Shiyu Ying** [1]

1   School of Economics, Hangzhou Normal University, Hangzhou 311121, China
2   School of Public Affairs, Zhejiang University, Hangzhou 310058, China
*   Correspondence: hongyunhan@zju.edu.cn; Tel.: +86-135-7578-6597

**Abstract:** The prevailing informal contracts of farmland transfer in China are facing frequent disputes and defaults, which call for effective self-enforcement mechanisms operating through transactors' reputations and social networks. However, the effects of reputation on contract choice and self-enforcement have not been thoroughly considered and examined by existing research in the case of farmland transfer. This study explores the reputation's ex-ante signaling effect on farmers' contract choices and the ex-post penalty effect on farmers' performance in informal contracts. Based on 403 transfer contracts obtained from a field survey conducted in the Hebei province of China, we apply the multinomial logit model and Heckman probit model to perform empirical analysis. The results show that, affected by the penalty effect, farmers with good reputations are more likely to fulfill informal contracts to avoid reputation damage and the resulting loss of future trading opportunities. However, in the ex-ante stage of contract choice, a farmer's reputation has no significant signaling effect on the formation of informal contracts. The informal contracts are chosen due to farmers' trust in the close social network and the demand for reduced transaction costs. These findings highlight the importance of personal reputation serving as a form of relational governance in the self-enforcement of informal contracts, which provides a means of enhancing the informal contract's effectiveness in terms of farmland transfer in the rural acquaintance society. It also provides insights into the necessity of creating a supportive environment for informal rules. Policies should encourage the building of personal reputation and establishment of good social norms to form a long-term, stable and reasonable contractual relationship for farmland transfer.

**Keywords:** reputation effect; social network; self-enforcement of informal contract; farmland transfer





## 1. Introduction

In the process of promoting rural revitalization in China, farmland transfer is critical for achieving scale operation and further stimulating the transformation of modern agriculture [1]. Given the complexity and uncertainty of the farmland rental transactions, an appropriate contract arrangement provides an institutional basis for ensuring the stability and reliability of farmland transfer [2–4]. In the case of farmland transfer in China, formal written contracts and informal oral contracts have a long-term coexistence [5,6], and informal contracts even account for a larger proportion due to the imperfection of the farmland rental market [7,8]. However, temporary and unstable informal contracts bring frequent disputes and defaults, which in turn result in the low effectiveness of contracts and further market disorder [2,8]. According to the China Rural Management Statistical Annual Report, the annual average disputes and defaults in farmland transfer in China totaled 417,999 in 2015–2018, involving the breach of contract rents, violation of the leasing duration, and interference with land use [9,10]. It is highly necessary to devise an effective self-enforcement mechanism to ensure the stable and orderly development of the market of farmland transfer.

The issue of informal contract arrangement and enforcement has attracted considerable interest. Previous studies attached importance to formal contracts, attributing farmers' preferences for informal contracts to the imperfect farmland tenure and compliance with traditional customs [11–13], and for the sake of convenience [10]. Beyond this, a strand of literature began to rethink the positive role of informal contracts in farmland transfer via the framework of transaction costs. Formal contracts often impose higher transaction costs for dispersed small farmers in the current imperfect farmland rental market [14]. Subsequently, studies focus on the enforcement of informal contracts of farmland transfer, thus illustrating the adaptability of informal contracts to the current rural system [15–17]. Informal contracts rely on self-enforcement mechanisms, in which no third party intervenes (e.g., courts) to judge whether a breach has taken place or to evaluate the damages resulting from the breach [18]. The specific self-enforcement mechanism plays a vital role in assuring informal contractual performance [19–21], including mutual trust [22,23], social capital [24,25], and farmers' demographic characteristics [26–29] on contractual performance.

In particular, reputation is a private form of capital in an informal contractual relationship and has a long-term effect on improving the contract's effectiveness [19]. Reputational capital is claimed as the key mechanism of informal contract self-enforcement [18]. On one hand, reputation has a signaling effect on facilitating contract establishment. Demonstrating a good personal reputation releases a signal that the trader is credible. This can help expand the scope of transactions, reduce transaction costs, and increase expected returns, thus improving the other party's confidence in the trader's performance and ultimately reaching a flexible contract. On the other hand, reputation is claimed as an invisible guarantee for the self-enforcement of the informal contract [30]. By the devaluation of the defaulter's reputational capital through the transmission of credit information, which results in their potential future losses in repeated games, informal contracts can be self-enforced [18]. This reputation effect based on private penalties becomes the key to contract self-enforcement. The effect of reputation, as an important determinant of informal contract choice and self-enforcement, has attracted increasing attention in empirical research. The majority of studies highlight the expected penalty effect of reputation on a trader's performance in a contractual relationship, focusing on the issue of implicit incentives on a CEO's performance [31,32], farmers' performance in private lending contracts [29,33,34], and enterprise performance in agricultural product contracts and financial contracts [35,36]. In the case of farmland transfer, a few studies have carried out theoretical analyses on the impact of farmers' reputations on their contractual performance in farmland transfer. For example, Liu and Lv [37] constructed a game model to analyze the conditions of contract enforcement in farmland transfer and verified that the consideration of reputation would expand the scope of contract self-enforcement. Hong and Qian [38] examined the effect of reputation on farmers' performance in oral contracts based on surveyed data. They used the evaluation as "high or low" from other villagers as a proxy for a farmer's reputation and found that farmers with a higher evaluation of reputation are more likely to fulfill the contract.

To clarify the effect of reputation is the key to ensuring the long-term effectiveness of informal contracts in farmland transfer. However, the effects of reputation have not been thoroughly considered and examined by existing research in the case of farmland transfer. Transactions in rural China are based on customs and relational rules, forming repeated games in a relatively closed environment. Therefore, informal rules may be more effective than formal rules in rural societies [5]. The long-term implicit effect of reputation on contract arrangement and the enforcement of farmland transfer merits close attention. There are still research gaps in explaining reputation's effects throughout the whole process of contract arrangement and enforcement. Firstly, there is a lack of a theoretical framework of reputation's effect incorporating the signaling effect and penalty effect of reputation. Few studies have distinguished the ex-ante and ex-post effects of reputation on contract parties. This is partially due to the fact that the ex-ante contract choice and the ex-post contract performance of farmland transfer are mostly treated as two separate parts, which

failed to provide a coherent explanation of the source of informal contracts' effectiveness. In other words, a farmer's behavior in choosing an informal contract and providing good performance to fulfill it may be commonly affected by reputation and other common variables. Correspondingly, beyond the preliminary research on the determinants of contract form choice and enforcement, there is no sufficient empirical examination of the mechanism of reputation's effect. As for the two specific effects of reputation, the signaling effect of reputation is mostly ignored in existing research, but it is precisely one of the most important determinants of establishing deals in rural areas. The penalty effect of reputation on contractual performance has attracted increasing attention in empirical research. However, the conditions under which the penalty effect will work need to be clarified further. Reputation may interact with other relational rules, such as rural social networks, and jointly affect farmers' choices and performance. The role of reputation played and its interactions with other informal rules need further examination.

This study contributes to the existing literature in three ways. Firstly, we investigate the mechanism through which personal reputation functions in the contract choice and self-enforcement by clarifying the ex-ante signaling effect and ex-post penalty effect. This provides insights into clarifying the role of reputation as an implicit form of private capital in improving contract effectiveness. Secondly, our study empirically examines the reputation effect in two stages, including the signaling effect in the stage of trading partner searching and the penalty effect in the stage of informal contract self-enforcement. We further explore the interaction between reputation and social network, another important self-enforcement mechanism in rural areas, and its effect on contract self-enforcement. This reveals the condition of the penalty effect embedded within the tight social networks in a rural society. The third contribution is the consideration of the possible endogeneity from the sample selection bias in our empirical strategy. Since we can only observe the performance of farmers who have chosen informal contracts, we apply the Heckman probit method to solve the problem of the selected sample.

The rest of this study is organized as follows. Section 2 constructs the theoretical framework and literature review. Section 3 introduces the materials and methods, including the study sites, data source, variables, and models. Section 4 presents the empirical results and discussion, and limitations are included. Section 5 concludes and provides policy implications.

## 2. Conceptual Framework

A conceptual framework of the effect of reputation on the contract arrangement in farmland transfer is constructed (Figure 1). Taking the point of signing the contract as the benchmark, we divide the contract arrangement process into two stages: ex-ante contract choice and ex-post contract enforcement. Our focus is on the choice of informal contracts and the effectiveness of self-enforcement mechanisms. Subsequently, we present how personal reputation may affect informal contract choice through the signaling effect and affect informal contract self-enforcement through the penalty effect.

### 2.1. Informal Contract and Self-Enforcement Mechanisms

The contract arrangement and its effectiveness are always an important issue in contract economics. Ex-ante contract choices and ex-post enforcing mechanisms together constitute a holistic system of contract arrangement. It provides the specific distribution of bilateral rights and responsibilities and the allocation of resources and risks, which serves as a governance method and matches with specific transactions [39,40]. Contracts are essentially classified into formal contracts and informal contracts in terms of the differences in form, completeness of terms, and enforcing mechanisms. A formal contract is usually written and has legal validity, and will be enforced by a third party, which is compulsory [41]. However, the complete contractual specification cannot be achieved due to the terms being unobservable beforehand and unverifiable afterward, which makes the negotiation and court enforcement costly [42,43]. An informal contract is a relational governance pattern

based on social ethics and cooperative relationship in repeated games [44] as a beneficial supplement to a formal contract by reducing transaction costs and improving contract adaptability [45,46]. More importantly, informal contracts have a long-term effect on individual behavior since they are based on moral constraints and mutual trust. They provide an invisible driving force for the improvement of social governance effectiveness [47,48].

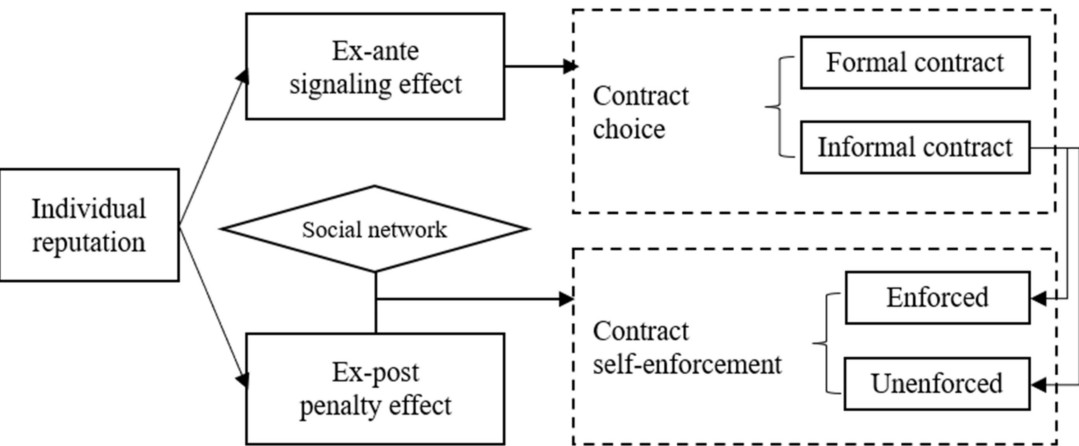

**Figure 1.** Theoretical framework of reputation effect on contract choice and self-enforcement.

In contrast to the court enforcement of formal contracts, informal contracts are mainly enforced by self-enforcement mechanisms [49], which operate by threatening the termination of the contractual relationship in the event of defaults [19]. In a self-enforcing contract, each party decides unilaterally whether to continue or end the contractual relationship. They make the decision by calculating whether the gain from breaching the contract is greater or less than the loss of future benefits from the termination of the agreement by the other party [18,19]. Traders will compare the short-term gains achieved from defaults or shading performance with the possible loss of the expected future profit stream if the contractual relationship is terminated for their defaults [30]. Performance is assured when the expected future loss is larger than the short-term gains. A self-enforcement mechanism is based on the rules of the relational governance of reputation [30], trust [50], and reciprocity [51]. A closer relational network can enhance the effectiveness of these relational governance measures [18]. A self-enforcement mechanism reduces the cost of contract enforcement by eliminating the negotiation cost of writing detailed contract terms and the time delay of court sanctions due to breaches [19].

## 2.2. Effect of Farmers' Reputation on Contracts in Farmland Transfer

Reputation as a type of intangible private performance capital is one of the most important determinants of contract arrangement. As Klein [19] stressed that sufficient reputational capital will cause transactors to rely on self-enforcement rather than court enforcement. Taking the point of signing a contract as the benchmark, the role of reputation in contract arrangements can be divided into the signaling effect and the penalty effect.

### 2.2.1. Ex-Ante Signaling Effect on Contract Choice in Farmland Transfer

Reputation can be an effective signal that conveys a trader's intrinsic characteristics and past behavior, which influences the expectations and final decisions of the parties involved [18,52,53]. Pioneering research on the lemon market by Akerlof [54] emphasized that reputation could reveal important information for resolving adverse selection under information asymmetry. Subsequently, extensive literature has affirmed the signaling function of reputation [52,55,56]. Firstly, reputation has the function of information disclosure. Kreps and Wilson [30] define reputation as the reflection of a personal historical record and utility function. Under the condition of information asymmetry, individual behavior in a market is not completely observable. A good reputation then is equivalent to an implicit

guarantee [57]. A trader's reputation, representing information about past behavior, can be a signal of their intrinsic traits and future performance, which can be displayed to other parties for judging their credibility [53]. Secondly, reputation has the function of signal delivery. Reputation information is exchanged and disseminated among various stakeholders, forming a reputation flow or network [58]. It lowers the information acquisition costs and increases information transparency, thus reducing adverse selection and moral hazards caused by information asymmetry [59,60]. Therefore, reputation flow can partially replace formal legal contracts and enhance market efficiency [58].

Based on that, the signaling effect of reputation operates through disclosing and delivering the information of contract parties and then forms an implicit constraint. It helps traders to expand the scope of the transaction, identify trustworthy counterparties and reduce transaction risks [61]. Therefore, regulations by reputation could be an effective complement to written contractual terms. As Kreps and Wilson [30] pointed out, regulations can be appropriately relaxed for a party with a high reputation, thus, reaching a loose and flexible contract. In the case of farmland transfer, a farmer's contract choice of informal contract in farmland transfer may be affected by the signaling effect of reputation. Given the information asymmetry of the transaction, a farmer's reputation is a key signal revealing their past and future behavior. A good reputation will convey the message that the farmer is trustworthy and allow the other party to be confident enough to relax the contract regulations. Even in transactions within acquaintance networks, farmers still rely more on reputation information over screening and choosing patterners. As Hong et al. [22] have found, farmers are more likely to enter into informal contractual relationships with reputable parties because a good reputation is more likely to generate mutual trust and relax the regulations against their opportunistic behavior.

### 2.2.2. Ex-Post Penalty Effect on Contract Self-Enforcement in Farmland Transfer

The ex-post penalty effect of reputation, which is also known as the incentive effect of reputation, is based on the reputation trading theory that defines personal reputation as a form of capital in the market. In the self-enforcement mechanism, defaulters may be faced with the penalty of reputational capital devaluation and the resulting possible loss of future trading opportunities [30]. Thus, transactors have the motivation to perform well in the contract to maintain a good reputation. Fama [62] firstly proposes that reputation is an implicit incentive, and the goal of establishing a reputation is to obtain long-term benefits. Kreps and Wilson [30] constructed the KMRW reputation model suggesting that the value of the reputation asset in repeated games motivates individuals to fulfill a contract. Given the incomplete information, both contract parties would supply cooperative behavior as long as the number of repeated games is sufficient [30]. In an informal contractual relationship, reputation is a type of private device providing incentives that ensure contractual performance [63]. The demand for maintaining a reputation imposes implicit self-restraint on hidden behaviors and increases the cost of defaulting, thus promoting self-enforcement [64]. In the transactions of farmland transfer, farmers value the construction and accumulation of their reputation, since a good reputation helps them to maintain trading partners and obtain long-term gains. Therefore, they will provide good performance in a contract. Huang [65] further defines reputation as a form of social capital, suggesting that a poor reputation represents an informal "social penalty", which makes breaking contracts a costly option [66].

Additional conditions are required for the penalty effect of reputation: breaching behavior can be detected by the partners in time and the counterparty can impose an effective punishment for the breach [43]. Based on the KMRW model, researchers extended Doe's theorem in the standard repeated game to the infrequent random matching games [67,68]. They verified that word-of-mouth messaging ensures that the reputation mechanism works effectively as long as the community is not too large to deliver information about disputes and defaults. Therefore, rapid and low-cost information delivery within the community social network is the prerequisite for the penalty effect of reputation [69]. A social network is

characterized by an identity within a small group, shared social norms, and mutual support based on reputation and trust [70]. In rural China, the differential social networks based on kinship and geo-relationships formed the "pattern of difference sequence" [71]. This becomes a network for information delivery [5] and construction. In the social networks of acquaintance societies, the consequences of defaulters' reputational damage could be more severe [72].

## 3. Materials and Methods

### 3.1. Study Sites and Data

The study site is Chengde City located in northeastern Hebei province, China, with a total area of 39,500 km$^2$ and a total population of 3,729,600, including a rural population of 2,797,704,000 rural population. There are 35,891,436,000 mu of agricultural land, including 4,396,761 mu of arable farmland. Given the rapid development of modern agriculture, Chengde has swiftly promoted farmland transfer. As of the end of 2014, the total transferred area was 833,481 mu (accounting for 19.5%) involving 130,000 farmer households (accounting for 15%), which is at the average level in Hebei province. A total of 78,200 written contracts were signed as of 2014, and numerous oral contracts existed[1]. Farmland transfer in Chengde also faces the problem of disputes and defaults. According to the statistics of the Agricultural Economic Statistics Report from the Chengde Municipal Bureau of Statistics, there were more than 800 disputes and defaults in Chengde City in 2010–2014. Problems such as rent arrears and farmland overuse by renters, requests for increasing rent, and the repossession of farmland are difficult to solve.

The data were collected by field survey by the research group from Zhejiang University in Chengde City, Hebei province in 2015. We chose five counties in Chengde City with a relatively higher transfer rate to obtain more information about transfer contracts, namely Shuangluan District, Chengde County, Fengning County, Luanping County, and Pingquan County (Figure 2). The questionnaire survey was conducted in 25 villages randomly chosen from these five counties. We randomly chose farmers as interviewees and performed face-to-face interviews. A total of 466 valid questionnaires were obtained after eliminating six questionnaires with missing information. The validity rate was 98.7%. After deleting the data of 63 samples of farmer households without farmland transfer, the sample of this study included 403 farmland transfer contracts from 287 farmer households. Some farmers may have had more than one farmland contract.

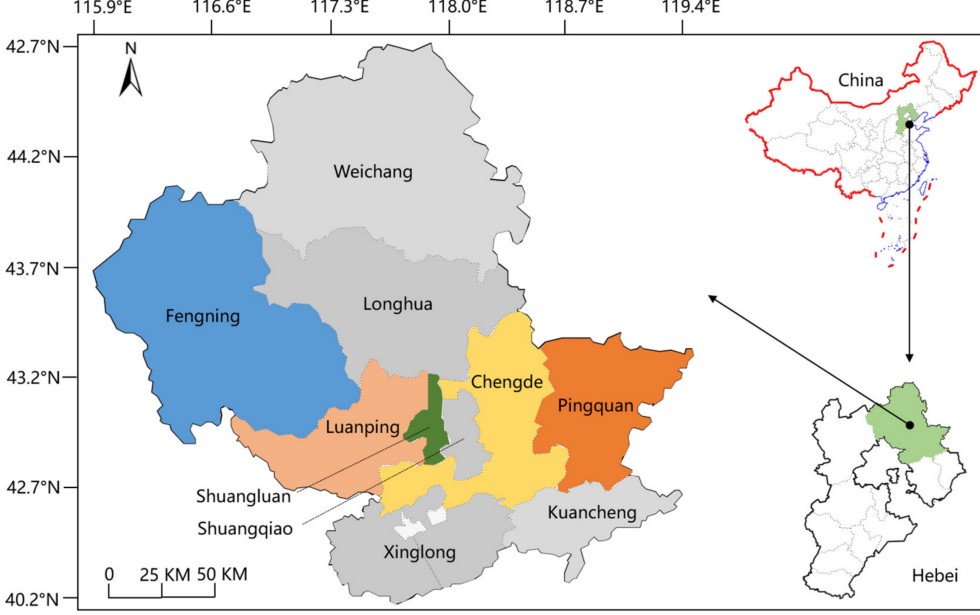

**Figure 2.** Study sites of Chengde City, Hebei province in China.

*3.2. Variable Selection*

3.2.1. Dependent Variables

*Contract form.* In the model of contract form choice, the dependent variable is the contract form. We take two classifications of contract form. Binary classification of a formal written contract (contract form = 0) and informal oral contract (contract form = 1) is taken as a baseline analysis, which is generally applied in the previous literature. Considering the complexity of real farmland rental transactions, we further divide the contract form into four categories to reveal the differences in inherent completeness, which are (1) oral contract only agreed upon by bilateral parties (OB); (2) oral contract witnessed by a third party (OT); (3) written contract drafted by both parties (WD); (4) unified written contract made by local government (WU). These contract forms are typical in our study sites. With the contracts witnessed by a third party and the transitions from oral contracts to written contracts, the completeness of contracts is gradually enhanced. The detailed definition and frequency of contract forms are shown in Table 1.

**Table 1.** Classification of contract forms and descriptive statistics.

| Contract Form | | Definition | Fre. | % |
|---|---|---|---|---|
| Informal oral contract | (1) Bilateral oral contract (OB) | Informal oral contract agreed by both sides, without the witness of a third party | 62 | 15.38 |
| | (2) Witnessed oral contract (OT) | Informal oral contract agreed by both sides and witnessed by a third party | 55 | 13.65 |
| Formal written contract | (3) Drafted written contract (WD) | Formal written contract (or brief notes, pledge, etc.) drafted by both sides | 148 | 36.72 |
| | (4) Unified written contract (WU) | Formal written contract provided by government departments or enterprises | 138 | 34.25 |
| Total sample | | - | 403 | 100 |

Data source: Survey of farmland transfer contract arrangement in Chengde City, Hebei province in 2015.

*Contract self-enforcement.* In the analysis of informal contract self-enforcement, the dependent variable is whether the informal contract is enforced, which can be represented by a binary choice of "whether there were any disputes or breach during the contract (no = 1; yes = 0)", referring to previous studies [73]. Since a farmer's contractual performance in fulfilling the contract is difficult to ascertain from continuous observation and measurement, we take the emergence of disputes and defaults as a proxy for contract self-enforcement.

3.2.2. Explanatory Variables

*Farmer's reputation.* Since it is difficult to directly measure the reputation level, other "soft information" about farmers can be applied to reveal their reputation situation, such as social capital, roles and relationships, and lending and borrowing history [33,66]. Referring to Lei and Li [35], using the records of enterprise historical bad behavior to represent enterprise reputation, we define a farmer's reputation as a continuous variable as the number of times that the householder mediated conflicts among other villagers in a year. The more times they mediated conflicts, the higher the farmer's reputation level. Practical experience of rural society shows that if there are contradictions and disputes between the two sides that cannot be reconciled, most farmers will invite individuals with a high social status to mediate [74]. We applied neither self-evaluation nor a dichotomy dummy proxy to evaluate farmers' reputation, which have been mostly applied in previous studies. The former cannot avoid the subjective cognitive bias [33], and the latter cannot reveal the difference in reputation level continuously and accurately [38].

*Social network.* The social network is denoted as the relationship between contracted parties. Referring to the concept of the "pattern of difference sequence" in rural China proposed by Fei [71], different types of social networks form a "concentric circle" around individuals. The tightness of a social network decreases from close kinship to distant kinship and also decreases with increased geographical distance increase. Referring to

the empirical strategy of Luo et al. [14], Hong [22], and Qian and Qian [24], we define social network as a multinomial dummy with four ordered categories: parents and close relatives (social network = 1), neighbors in the same village (social network = 2), farmers from other villages (social network = 3), and agricultural business entities such as large family farms, cooperatives, and agricultural enterprises (social network = 4). The tightness of the social network decreases in order. A closer relationship means stronger relational rules embedded, which forms a stronger impact on farmers' behavior. Hence, farmers are more likely to fulfill the contracts with acquaintances because of trust, loyalty, and an invisible binding force. Ties between strangers are the weakest, where the relational governance rules are not valid enough to support contract self-enforcement.

### 3.2.3. Control Variables

Referring to Liu et al. [8], Luo et al. [14], and Chen and Wang [24], a series of variables are selected to control the effect on contract form choice and contract self-enforcement, including the characteristics of farmland, characteristics of the contract, household agricultural operation and farmer households' demographics. Table 2 provides the definitions of all the variables.

**Table 2.** Definition of variables.

| Variables | Definition of Variables |
|---|---|
| **Dependent variables** | |
| Contract form | Binary: 1 = informal contract, 0 = formal contract<br>Multinomial: 4 discrete categories of oral contract only agreed by bilateral parties (OB), oral contract witnessed by a third party (OT), written contract drafted by both parties (WD), and unified written contract made by local government (WU) |
| Contract self-enforcement | 1 = contract self-enforced, no disputes or defaults<br>0 = contract unenforced |
| **Independent variables** | |
| Farmer's reputation | Number of times that the householder mediated conflicts and disputes among other villagers in a year |
| Social network | Contracting parties are parents and relatives = 1; neighbors = 2; farmers from other villages = 3; cooperatives, agricultural enterprises and other subjects = 4 |
| **Control variables** | |
| Transferred area | Area of transferred farmland (mu [a]) |
| Transfer rent | Payment of transferred farmland [b] (CNY/mu) |
| Transfer modes | Organized transfer = 1; spontaneous transfer = 0 |
| Contract duration | Clearly agreed duration in contract = 1; otherwise = 0 |
| Rent prepayment clause | Clause stipulating that the contract rent should be paid at the beginning of the transferred year = 1; no such clause = 0 |
| Rent adjustment clause | Clause stipulating that the contract rent should rise with the crop market price = 1; no such clause = 0 |
| Agricultural durable assets | Total value of household-owned agricultural machinery and large livestock (thousand CNY) |
| Proportion of agricultural income | Ratio of household agricultural income to total income (%) |
| Information access | Total fee of communications, network, express and TV annually (CNY) |
| Number of transfer contracts | Total number of contracts of farmland transfer |
| Farmer's trust | Concern about breach by the other party = 1; otherwise = 0 |
| Big clan member | Householder is a member of big clan in the village = 1, otherwise = 0 |
| Age of householder | Age of householder (Years) |
| Education of householder | Years of education of householder (Years) |

Notes: Based on the survey period, price information including the transfer rent, value of assets, incomes, and expenditure are all presented as the level in 2014. [a] Mu is a unit of farmland scale in the Chinese system of measures. For the consistency of the measurement in CFPS, we applied mu as the unit of farmland scale in this study. 1 mu = 0.067 hectare, the same as below. [b] Transfer rent is reflected by the amount of actual cost for transferred farmland in 2014. The rent paid by crops is converted into cash according to the market price of crops in 2014, and the share of rent is subject to the actual amount of cash obtained.

- Characteristics of farmland transfer. The variables of *transferred area*, *transfer rent*, and *transfer mode* are included to reveal the transferred plot characteristics. It should

be noted that *transfer modes* are divided into spontaneous transfer and organized transfer, and the latter may be organized by the village and often matches with written contracts.

- Characteristics of contracts. The variables of *contract duration*, *rent prepayment clause*, and *rent adjustment clause* are included to reveal contract characteristics. Clear and appropriate design of contract clauses will lead to a farmer's strong expectation for sustainable benefits over a long period and may promote their self-enforcement [18]. *Contract duration* denotes whether there is a clearly agreed upon duration in the contract. The *rent prepayment clause* is a clause stipulating that the rent should be paid at the beginning of the contract period. The *rent adjustment clause* is a clause stipulating that the rent should rise with the crop market price. According to our survey in the study area, the rent adjustment clause is used to protect farmers' benefits and reduce market risks. An example of such a clause is, "if the price of the crop rises in the future, the rent of transferred farmland should increase annually (or every three or five years) according to the market price; if the price drops, the rent will remain unchanged".

- Household agricultural operation. Household agricultural operations relying on farmland will affect a farmer's choice and performance in farmland transfer. The variables of *agricultural durable assets* and *proportion of agricultural income* are included. *Agricultural durable asset* represents household asset specificity in agricultural production. Higher asset specificity may have a lock-in effect on farmers. It causes them to expect the contract to be effective to protect their asset investment and future gains. *Proportion of agricultural income* represents a farmer's dependence on agricultural production. It is supposed to prompt farmers to choose formal contracts with stronger validity. Even in an informal contract relationship, the dependence may also promote farmers to provide ex-post good performance to enforce the contract in order to recover the farmland after the contract's expiration.

- Farmer households' demographics. In this part, the household characteristics and farmer householder's characteristics are both included. Variables of household characteristics include *information access* and *number of transfer contracts*. *Information access* reflects a farmer's information accessibility measured by the total fee of communications, network, express and TV, which is supposed to affect contract cognition and choice preference. *Number of transfer contracts* is represented by the total number of farmland transfer transactions. The learning effect and experience accumulation in multiple transactions may have an impact on their preferred contract form [75]. The farmer householder's characteristics include *farmer's trust, big clan member, age of householder*, and *education of householder*. *Farmer's trust* is defined as whether farmers are concerned about the reputation of the other party when signing the contract. If the reputation level of the other party is not sufficient to convince the farmer, they may tend to choose complete contracts to protect themselves. In the process of contract enforcement, concern for the reputation of the other party will also affect the performance of their own behavior. *Big clan member* indicates that the farmer householder is a member of a big clan in the village, namely having a tighter genetic relationship and greater social capital. A farmer who is a member of a big clan is typically the original resident of the village and has more relatives with strong ties in the village; thus, he can obtain more reciprocal benefits.

### 3.2.4. Descriptive Analysis

Descriptive analysis and the results of the *t*-test for group means are shown in Table 3. The overall enforcement rate of farmland transfer contracts is at a relatively high level (87.3%). Farmers present more than one instance of mediating conflicts, and their transfer contracts are more likely to be signed by other villagers and agricultural business entities. The average transferred area is 5.732 mu with an average rent of 679.326 CHY/mu. Only 34.5% of the farmland transfer is organized by the village. Only 54.2% of contracts set a clearly agreed duration, 77.4% include a rent prepayment clause, and 4.7% of con-

tracts include a rent adjustment clause. On average, farmer households have more than one transfer contract. Farmer households own 2413 CHY of agricultural durable assets on average, and 24.9% of their income is derived from agricultural operations. The average fee for information access is 824.072 CHY. The average age of farm householders is around 50 years old, with 7 years of education, and 59.8% of the householders are big clan members. Among the total sample, 74.7% of the householders do not trust the other contract party and are concerned about breaches of contracts.

**Table 3.** Descriptive statistics of contract form selection and performance mechanism variables.

| Variables | Total Sample | | Oral Contract Sample | | Written Contract Sample | | *t*-Test |
|---|---|---|---|---|---|---|---|
| | Mean | SD | Mean | SD | Mean | SD | |
| Contract self-enforcement | 0.873 | 0.333 | 0.786 | 0.412 | 0.909 | 0.288 | −0.123 *** |
| Farmer's reputation | 1.117 | 4.208 | 0.991 | 3.010 | 1.168 | 4.613 | −0.176 |
| Social network | 2.700 | 1.164 | 1.274 | 0.551 | 3.283 | 0.781 | −2.010 *** |
| Transferred area | 5.732 | 9.601 | 5.202 | 3.939 | 5.949 | 11.115 | −0.747 |
| Transfer rent | 679.326 | 442.646 | 319.355 | 318.018 | 826.587 | 400.312 | −507.233 *** |
| Transfer modes | 0.345 | 0.476 | 0.085 | 0.281 | 0.451 | 0.498 | −0.366 *** |
| Contract duration | 0.542 | 0.499 | 0.085 | 0.281 | 0.730 | 0.445 | −0.644 *** |
| Rent prepayment clause | 0.774 | 0.419 | 0.265 | 0.443 | 0.983 | 0.131 | −0.718 *** |
| Rent adjustment clause | 0.047 | 0.212 | 0.085 | 0.281 | 0.031 | 0.175 | 0.054 ** |
| Agricultural durable assets | 2.413 | 11.814 | 2.844 | 9.227 | 2.236 | 12.732 | 0.608 |
| Proportion of agricultural income | 0.249 | 0.214 | 0.271 | 0.221 | 0.240 | 0.211 | 0.031 |
| Information access | 824.072 | 628.990 | 763.092 | 393.183 | 847.608 | 702.170 | −84.516 |
| Number of transfer contracts | 1.787 | 0.916 | 1.863 | 0.999 | 1.755 | 0.880 | 0.108 |
| Farmer's trust | 0.747 | 0.435 | 0.556 | 0.499 | 0.825 | 0.380 | −0.270 *** |
| Big clan member | 0.598 | 0.491 | 0.607 | 0.491 | 0.594 | 0.492 | 0.012 |
| Age of householder | 50.258 | 9.041 | 48.034 | 7.973 | 51.168 | 9.304 | −3.134 *** |
| Education of householder | 7.864 | 2.441 | 7.282 | 2.370 | 8.101 | 2.434 | −0.819 *** |
| N | 403 | | 117 | | 286 | | - |

Data source: Survey of farmland transfer contract arrangement in Chengde City, Hebei province in 2015. Note: ***, ** represent the significance of the mean difference between the oral contract group and the written contract group at levels of 1% and 5%, respectively.

Among the total 403 farmland transfer contracts, 117 are oral contracts, accounting for 29.1%. According to the results of the *t*-test, there are major differences between oral contracts and written contracts. The enforcement rate of written contracts (90.9%) is significantly higher than that of oral contracts (78.6%). The average reputation level of farmers who choose written contracts (1.119 times of conflict mediation) is slightly higher than that of oral contracts (0.991 times of conflict mediation) with no significant difference. Oral contracts exhibit significant informal features, as follows. (1) Oral contracts are more often signed among close friends and relatives; thus, the social network relationship between contract parties in oral contracts (1.274) is significantly closer than that in written contracts (3.283). (2) More than 90% of the oral contracts are in the spontaneously transfer mode and with no clear terms, which are significantly different from those of written contracts. (3) The average area transferred (5.202 mu) under oral contracts is lower than that of the written contracts (5.961 mu). (4) The average transfer rent (319.355 yuan/mu) is significantly lower than that of written contracts (827.733 yuan/mu). (5) The possibility of setting a rent prepayment clause in oral contracts (26.5%) is significantly lower than that for written contracts (98.3%), while the possibility of setting a rent adjustment clause in oral contracts (8.5%) is significantly lower than that for written contracts (3.1%). Based on the differentiation shown in the descriptive analysis, the determinants of farmers' preferences for informal contracts, and their performance in informal contracts need further empirical examination and discussion.

*3.3. Estimation Models*

3.3.1. Model of Contract Form Choice

One of the main purposes of this study is to investigate farmers' choices of contract form in farmland transfer. The basic model is as follows.

$$Form_i = \alpha_0 + \alpha_1 Reputation + \alpha_2 X + \mu_1 \tag{1}$$

where, $Form_i$ is the dummy variable representing different contract forms. *Reputation* is the farmer's reputation level. $X$ is the vector of explanatory variables, including characteristics of farmland, household agricultural operation, and farmer households' demographics. $\alpha_1$, $\alpha_1$ are the coefficients to be estimated. If the parameter $\alpha_1$ is significantly positive, it indicates that farmers' reputation significantly promotes the choice of informal contracts, namely there is a significant signaling effect. $\mu_1$ is the random error term. Firstly, we apply the binary logit model with a binary dummy $Form\_b_i$ as:

$$\Pr(Form\_b_i = 1 | X_i) = \frac{e^{X_i \beta}}{1 + e^{X_i \beta}} \tag{2}$$

where, $Form\_b_i = 1$ denotes informal contracts (OB and OT), and $Form\_b_i = 0$ denotes formal contracts (WD and WU).

Further, we expanded the model by dividing oral contracts and written contracts into sub-categories. Here, $Form\_m_i$ is the dummy variable representing four categories of contract form (as shown in Table 1). To reveal the categorical choice among heterogenous contract forms, we adopt the multinomial logit model (Mlogit) as follows:

$$\Pr(Form\_m_i = j | X_i) = \frac{e^{X_i \beta_{j|b}}}{1 + \sum_{j=1}^{j} e^{X_i \beta_{j|b}}}, \ j = 1,\ 2,\ 3,\ 4 \tag{3}$$

where $\Pr(Form\_m_i = j | X_i)$ is the probability that farmer $i$ selects contract form $j$, and $b$ is the base outcome (in this study the form WU with the highest completeness).

3.3.2. Model of Informal Contract Enforcement

Based on examining the determinants of farmers' choice of informal contracts, we further explore the enforcing mechanism of these informal contracts. To examine the penalty effect of reputation on the performance of informal contracts, we take farmers' reputation, social network, and their interaction effect as the core explanatory variables. The baseline regression model is as follows:

$$\begin{aligned} Enforcement_i = \gamma_0 + \gamma_1 Reputation + \gamma_2 Snetwork + \gamma_3 Reputation \times \\ Snetwork + \gamma_4 X_i + \varepsilon_1 \end{aligned} \tag{4}$$

where, $Enforcement_i$ is the binary variable representing whether the contract is enforced. $Enforcement_i = 1$ if no dispute or breach of contract, which means the contract is effectively enforced, $Enforcement_i = 0$ otherwise. $Reputation_i$ is farmer's reputation level. $Snetwork_i$ is a discrete variable that denotes the social network. $X_i$ is the vector of other explanatory variables, including characteristics of farmland, characteristics of contracts, household agricultural operation, and farmer households' demographics. $\gamma_1 \sim \gamma_4$ are the parameters to be estimated, $\varepsilon_1$ is the random error term.

We firstly take a binary probit model as the base model. Further, the possible endogeneity derived from the sample selection bias needs to be checked [76]. In the examination of informal contract enforcement, we can only observe the situation of farmers who choose informal contracts in farmland transfer beforehand. This results in a non-random sample set, which is selected based on the condition of "choosing an informal contract in farmland transfer". There may be some common variables affecting contract choice and

self-enforcement. Hence, we apply the Heckman probit model to deal with the possible endogeneity problem of sample selection bias.

In the Heckman probit model, the selection equation is set by a binary variable $Informal_i$, which is "whether farmers choose informal contract (1 = yes, 0 = no)". The model of contract enforcement can be generalized as Equation (5), showing that the contract enforcement is conditional on a given probability of the farmer's choice of enforcement:

$$Enforcement_i = \begin{cases} \text{Observable, if } Informal_i = 1 \\ \text{Unobservable, if } Informal_i = 0 \end{cases} \tag{5}$$

where, $Enforcement_i$ is whether the contract is enforced, $Inforcement_i$ is the binary variable denoting the condition of choosing informal contracts (1 = informal contract, 0 = formal contract). The selection equation of the Heckman probit model is set as follows:

$$Informal_i^* = f(Z_i) + \varepsilon_2 \tag{6}$$

$$Informal_i = \begin{cases} 1, \text{ if } Informal_i^* > 0 \\ 0, \text{ if } Informal_i^* \le 0 \end{cases} \tag{7}$$

where, $Informal_i^*$ is the latent variable determining whether a farmer chooses the informal contracts, $Z_i$ is the vector of exogenous determinants of a farmer's contract choice. Here, $Informal_i^*$ is a function of $Z_i$, $\varepsilon_2$ is the random error term. Then,

$$\varepsilon_1 \sim N\left(0, \sigma^2\right)$$

$$\varepsilon_2 \sim N(0, 1)$$

$$corr(\varepsilon_1, \varepsilon_2) = \rho$$

where, $\rho \neq 0$ implies that the random error terms of the two equations of contract enforcement and contract selection are correlated, which means the Heckman probit model is necessary to solve the problem of sample selection bias.

## 4. Empirical Results and Discussion

### 4.1. Results of Contract Form Choice Model

The empirical analysis of contract choice provides evidence of the signaling effect of reputation. The independence of the irrelevant alternative (IIA) test is a prerequisite of the Mlogit model to verify the assumption that the relative likelihood of choosing category A over category B will not change if a third alternative is present [77]. We use the Hausman-McFadden test and the IIA assumption ($H_0$: Odds (outcome-j vs. outcome-k) are independent of other alternatives) and the results show that $H_0$ cannot be rejected for each category (see Table A1 in Appendix A). This indicates that the Mlogit model is appropriate. The VIF test is also applied, and the results (mean VIF = 1.6) verify that there is no multicollinearity of variables (see Table A2 in Appendix A). The regression results of the baseline binary logit model and Mlogit model are shown in Table 4. We also apply the Mprobit model with the same set of variables to check the robustness (see Table A3 in Appendix A). To focus on the explanation of farmers' choice of informal contract, we take the alternative of the unified written contract (WU) as the base category of the Mlogit and Mprobit model. Considering the possibility that one farmer may sign more than one transfer contract, we adopt clustering robust standard deviation to eliminate the bias. Comparing the results of coefficients and significances in the binary logit model, Mlogit model, and Mpobit model, the variables are generally consistent, indicating the forces that drive farmers to choose informal oral contracts (OB and OT) are generally consistent, indicating the robustness of the empirical strategy. The following detailed explanations are based on the Mlogit model results.

**Table 4.** The results of contract form choice model.

| Variables | Binary Logit (1 = Informal) | Mlogit (Based on WU) | | |
|---|---|---|---|---|
| | | (1) Y = OB | (2) Y = OT | (3) Y = WD |
| Farmer's reputation | −0.093 | −0.092 | −0.164 ** | −0.057 |
| | (0.0667) | (0.085) | (0.083) | (0.070) |
| Social network | −2.437 *** | −2.553 *** | −2.566 *** | −0.489 |
| | (0.448) | (0.546) | (0.529) | (0.333) |
| Transferred area | −0.110 ** | −0.097 * | −0.084 * | 0.008 |
| | (0.0497) | (0.058) | (0.049) | (0.023) |
| Transfer rent | −0.003 *** | −0.001 | 0.000 | 0.005 *** |
| | (0.001) | (0.001) | (0.001) | (0.001) |
| Transfer modes | −1.055 | −2.792 *** | −1.958 ** | −2.434 *** |
| | (0.714) | (0.908) | (0.772) | (0.626) |
| Agricultural durable assets | −0.034 *** | −0.041 ** | −0.051 ** | −0.017 |
| | (0.0115) | (0.019) | (0.020) | (0.024) |
| Proportion of agricultural income | 1.014 | 1.913 | 1.377 | 1.627 |
| | (0.912) | (1.196) | (1.124) | (1.084) |
| Information access | −0.0002 | 0.000 | −0.000 | 0.001 * |
| | (0.000) | (0.001) | (0.001) | (0.000) |
| Number of transfer contracts | 0.0122 | 0.017 | 0.017 | −0.136 |
| | (0.481) | (0.422) | (0.328) | (0.606) |
| Farmer's trust | −0.918 | −0.709 | 0.287 | 1.841 *** |
| | (0.586) | (0.698) | (0.666) | (0.451) |
| Big clan member | −0.869 | −0.684 | −1.252 ** | −0.164 |
| | (0.546) | (0.652) | (0.623) | (0.437) |
| Age of householder | 0.00181 | 0.035 | −0.068 * | −0.050 ** |
| | (0.0314) | (0.043) | (0.036) | (0.020) |
| Education of householder | −0.128 | −0.107 | −0.283 ** | −0.149 * |
| | (0.115) | (0.153) | (0.133) | (0.089) |
| Constant | 9.484 *** | 6.837 ** | 12.427 *** | 0.389 |
| | (2.533) | (3.139) | (2.747) | (1.841) |
| Observation | 403 | | 403 | |
| Pseudeo R2 | 0.741 | | 0.634 | |
| Prob > chi2 | 0.000 | | 0.000 | |

Notes: ***, **, * represent significance at level of 1%, 5% and 10% respectively; numbers in parentheses are cluster robust standard error.

Firstly, the improvement of a farmer's reputation significantly reduces the probability of choosing OT, and it has no significant effect on other incomplete contract form choices. This suggests that the ex-ante signaling effect of reputation on contract choice in farmland transfer is not significant but rather makes farmers less inclined to choose incomplete contracts. As supporting evidence, the information access also has no significant effect on the two categories of oral contracts. The farmer's reputation is not an effective signal to reveal their credibility through information disclosure and information delivery. For traders with farmers, a farmer's higher reputation is not convincing enough for them to sign a loose contract.

Rather than personal reputation, which signals a farmer's own credibility, the factors that influence contract choices are the farmer's expectations and judgments of the other party's performance under the constraints of informal rules. It is reflected by the results of *social network*, *big clan member*, and *farmer's trust*. Social network has a significantly negative effect on the probability of farmers choosing the two types of oral contracts, indicating that a closer relationship drives farmers to choose informal contracts. Social network represents a form of bilateral relational regulation in transactions, and it is an alternative to market rules. In closer relationships with families or relatives, farmers have less information asymmetry, less uncertainty, and easier supervision [5]. This builds up better trust among the farmer and the other contracting party and allows them to be more comfortable in choosing incomplete contracts. A big clan, as an implicit reflection of a farmer's social capital level, shows a

consistent effect on contract choice. The regression results of farmer's trust confirm this finding, which reflects farmer's expectations more intuitively. However, this effect is limited to enabling farmers to switch from WU to WD, which remains within the scope of formal written contracts. This suggests that the informal rules that rely on social capital are not sufficiently strong to make farmers confident enough to choose oral contracts.

Consistent with the majority of previous studies, reduced transaction costs are an important reason that farmers choose incomplete contracts with the results of *transferred area* and *agricultural durable assets*. These two variables have significantly negative impacts on the probability of farmers choosing the two types of oral contracts but have no significant effect on choosing the drafted written contract. This indicates that when the farmland scale is small and the agricultural durable assets input of transferred farmland is low, farmers are more likely to choose informal contracts to save on transaction costs. This is in line with previous studies [22]. In other words, low asset specificity means a lower risk of underlying asset value loss. It means that farmers feel no need to spend extra time and engage in tedious procedures to sign written contracts, which may specify clear compensation clauses [78]. Moreover, the positive effect of *social network* on incomplete contract choice also suggests that farmers express demands for contract flexibility to save on transaction costs. Since farmers regard farmland as a type of security, they may express a demand for the resumption of farmland as appropriate to manage risks. If the farmland is transferred among acquaintances, it is convenient to negotiate mutually or terminate at any time under an oral contract [79].

The intervention of the local government will also affect a farmer's contract choice. The results of *transfer modes* show that the probability of farmers choosing three types of incomplete contracts significantly increases under the spontaneous transfer modes—that is, farmers are more likely to sign a unified written contract under organized transfer. Organized transfer occurs mostly between one enterprise or cooperatives and multiple farmers, which is usually organized by the village collective. A unified formal contract will be supplied by the enterprise or local government. In such cases, the contract form is not the "choice" of farmers, but their "acceptance" of the unified contract arrangement. Moreover, combined with the results of *age of householder* and *education of householder*, it can be concluded that a farmer's human capital level affects the choice of contract form. Younger and less educated farmers possess relatively weak legal awareness and contract spirit and may not highly value the farmland. They are more likely to choose OT or WD out of convenience. In addition, it is worth noting that the transfer rent does not significantly affect a farmer's choice between an oral contract and a written contract, but compared with WU, it increases the probability of farmers choosing WD. A plausible reason is that the rent level is not determined by the market but is linked to the contracting party. As far as the phenomenon in our field survey is concerned, relatively high contract rent usually occurs not in the transactions with cooperatives and enterprises but in the transactions with the neighbors and local large family farms (Table 5). The cooperatives and enterprises involving multiple farmer households tend to supply lower rents due to the coordination of the local government. However, in order to obtain farmland from the farmers involved in the operation area, the large family farms will offer a higher competitive rent level when signing contracts. In this case, for the sake of flexibility of contract adjustment, both parties mostly adopt WD rather than WU.

**Table 5.** Descriptive statistics of rent levels of different transfer contract parties.

| Transferred Contracting Party | Average Rent (CNY/mu) |
| --- | --- |
| Parents and relatives | 298.04 |
| Neighbors | 699.46 |
| Farmers from other villages | 1177.93 |
| Cooperatives, agricultural enterprises and other subjects | 505.50 |
| Total sample | 678.19 |

Data source: Survey of farmland transfer contract arrangement in Chengde City, Hebei province in 2015.

Moreover, compared with the result of WD (column (3) in Table 4), there are highly consistent results for the two types of oral contracts (column (1) and (2) in Table 4). This means that the discrepancy between formal contracts (WU and WD) and informal contracts (OB and OT) is larger than the discrepancy among different levels of contract completeness. Although we define WD as a type of formal but incomplete contract, it is significantly different from the informal oral contract in terms of the farmer's choice. The main determinants of WD/WU choice are the farmer's trust and expectation, and the transfer modes and transfer rent jointly reflect the government's intervention in circulation. Meanwhile, the determinants of informal/formal contract choice are the social network with informal rules embedded and the possibility of reducing transaction costs. Generally, farmland transfer is a particular type of transaction embedded within the unique rural man–land relationship. It is not only focused on economic benefits but also needs to consider factors such as social relationships, rural customs, and past and future reciprocity [71,79]. Contracts with lower completeness can be an effective supplement to contracts with higher completeness through their lower transaction costs and higher contract flexibility.

### 4.2. Results of Self-Enforcement of Informal Contracts

The empirical analysis on informal contract self-enforcement provides evidence of the penalty effect of reputation. In Table 6, columns (2) and (3) refer to the binary probit model of contract enforcement with informal contract samples as the baseline. Columns (4)–(7) refer to the Heckman Probit model which includes the observation equation and the selection equation. The variables of *number of transfer contracts* and *transfer modes* are defined as exogenous variables of contract self-enforcement and not included in the observation equation. Multiple numbers of transfer contracts may be derived from many cases, including the transaction of multiple plots, or transactions at different time points for the same plot, which is exogenous to the occurrence of disputes or defaults. Transfer modes differentiating organized transfer and spontaneous transfer are determined by local policies of promoting scale transfer and investment by agricultural entities [73,80]. The coefficients and significances of the binary probit model and the observation equation of the Heckman probit model are basically consistent, indicating the robustness of our empirical strategy. The results of the Heckman model indicate that $\rho$ is significant at the 5% level, and the LR test shows that $\rho$ is significantly different from 0 (Chi2 (1) = 5279.344, Prob > chi2 = 0.000, $H_0$: $\rho = 0$ can be rejected). This suggests that sample selection bias does exist in the issue of the self-enforcement of informal contracts, and the Heckman model is appropriate to solve the endogeneity problem. The following interpretation of the empirical results is based on the Heckman probit model.

### 4.2.1. Penalty Effect of Reputation on Informal Contract Self-Enforcement

The regression results indicate that a farmer's reputation level has a significant impact on contract self-enforcement at the 1% level, which indicates that there is a penalty effect of reputation. On one hand, reputation provides an implicit incentive for farmers to spontaneously perform well in contract relationships [80]. Since farmers treat reputation as a type of social capital, on the premise of pursuing the maximization of their own long-term interests, they cannot ignore the probable future losses caused by the reputational damage. In order to maintain their reputation accumulation, farmers will display self-restraint and reduce opportunistic as well as uncertain behaviors. On the other hand, reputation is an internalized value system instilled by the community to its members [38]. The personal reputation of farmers is subject to the implicit supervision within the rural community. Mutual supervision causes farmers to abide by the common norms formed within the community and form stable long-term cooperation expectations.

**Table 6.** The results of informal contract self-enforcement model.

| Variables | Binary Probit (1 = Contract Enforced) | | Heckman Probit | | | |
|---|---|---|---|---|---|---|
| | | | Observation Equation | | Selection Equation | |
| Farmer's reputation | 3.973 *** | (1.504) | 3.341 *** | (1.144) | −0.067 | (0.045) |
| Social network | −3.194 *** | (0.675) | −3.030 *** | (0.515) | −1.278 *** | (0.183) |
| Farmer's reputation × social network | −1.856 *** | (0.729) | −1.576 *** | (0.559) | | |
| Transferred area | 0.040 | (0.051) | 0.038 | (0.045) | −0.071 ** | (0.028) |
| Transfer rent | 0.003 ** | (0.001) | 0.002 * | (0.001) | −0.002 *** | (0.000) |
| Contract duration | −0.260 | (0.805) | −0.205 | (0.464) | | |
| Rent prepayment clause | 0.989 | (0.663) | 0.801 *** | (0.297) | | |
| Rent adjustment clause | −7.793 *** | (2.097) | −6.110 *** | (1.345) | | |
| Agricultural durable assets | 0.187 ** | (0.080) | 0.167 ** | (0.073) | −0.020 *** | (0.007) |
| The proportion of agricultural income | −0.189 | (1.248) | 0.087 | (0.907) | 0.559 | (0.514) |
| Information access | −0.005 *** | (0.001) | −0.004 *** | (0.001) | −0.000 | (0.000) |
| Farmer's trust | −0.029 | (0.511) | −0.249 | (0.401) | −0.535 | (0.325) |
| Big clan member | 1.872 *** | (0.619) | 1.550 *** | (0.526) | −0.362 | (0.270) |
| Age of householder | −0.053 | (0.038) | −0.042 * | (0.024) | −0.006 | (0.015) |
| Education of householder | 0.164 | (0.165) | 0.090 | (0.132) | −0.096 * | (0.058) |
| Number of transfer contracts | | | | | 0.057 | (0.196) |
| Transfer modes | | | | | −0.583 * | (0.347) |
| Constant term | 8.742 *** | (2.896) | 7.693 *** | (2.246) | 5.821 *** | (1.249) |
| Observation | 117 | | 117 | | 403 | |
| Pseudeo R2 | 0.611 | | | | | |
| Prob > chi2 | 0.000 | | | | | |
| athrho | - | | 14.377 ** (5.890) | | | |
| LR test | - | | chi2 (1) = 5279.34, Prob > chi2 = 0.000 | | | |

Notes: ***, **, * represent significance at levels of 1%, 5%, and 10%, respectively; numbers in parentheses are cluster robust standard errors.

The interaction of a farmer's reputation and social network has a significantly negative effect on contract self-enforcement at the 1% level, indicating that a tighter social network enhances the penalty effect of reputation. The penalty effect of reputation in an informal contract operates by the transmission of credit information in the social network, which results in reputation damage for defaulters in the market. The devaluation of reputational capital is more effective under the conditions of symmetrical information, rapid information spread, and long-term frequent trading [78]. These are precisely the characteristics of rural acquaintance networks. Under the condition of relatively closed regions and fixed members, the behaviors such as concealment, deception, and other breach behaviors are easily developed and quickly spread among members within the group. Farmers will avoid dealing with defaulters in the future, thus forming an underlying penalty mechanism. A closer social network means a higher degree of information symmetry and more efficient dissemination, which helps to form a stronger penalty effect of reputational damage. Farmers who are afraid of losing the opportunity for future transactions and even interpersonal trust, an important protection mechanism, will perform well spontaneously. As Guo [5] claimed, the accumulation of trust in the tight social network makes the self-enforcement of small-scale farmland transfer contracts more effective.

Social network has a significantly negative impact on contract self-enforcement at the 1% level. The closer the relationship with the contracting party is, the higher the probability of contract self-enforcement is. The different enforcing mechanisms corresponding to the diversified contract party need to be guaranteed by the relational governance mechanism embedded in the social network. Under the "pattern of difference sequence" of the rural social network in China, the strongest relational ties between families and close relatives attach powerful validity to relational rules, which make farmers more likely to fulfill the contracts because of trust and loyalty. Meanwhile, cooperatives and enterprises are considered to be strangers, where the relational governance rules are not valid enough to motivate farmers to perform well.

4.2.2. Other Effects on the Self-Enforcement of Informal Contracts

Based on the results of other control variables, the effects on informal contract self-enforcement include the deterministic current incentives, the lock-in effect of asset specificity, and the compensation of farmers' social capital for human capital. The *transfer rent* and *rent prepayment clause* significantly increase the probability of self-enforcement, respectively at the 10% level and 1% level. Farmer's performance seeks both "survival rationality" and "economic rationality". The higher the income brought by farmland transfer, and the smaller the uncertainty of the income, the higher the possibility of a farmer's good performance [5,81]. The expected benefits of future cooperation can promote the current cooperation of parties [46]. Contrary to our expectations, the rent adjustment clause significantly reduces the probability of self-enforcement at the 1% level. One possible explanation is that, although the rent adjustment clause is designed to provide stable income expectations for farmers in long-term contracts, the lack of enforcement mechanisms in informal contracts makes it ineffective. The flexible clause in oral commitments cannot provide future income expectations but forms a kind of uncertainty, thus, it cannot provide an effective incentive for farmers to fulfill the contract. In addition, a promise to increase rent may cause farmers to have higher expectations of rent than the actual level, which is described as the "price illusion" by Luo and Liu [73]. When farmers overestimate the transfer rent level, they may engage in disputes or defaults due to dissatisfaction with the current rent level.

The lock-in effect of asset specificity on contract enforcement is reflected by the result of *agricultural durable assets*, which increases the probability of self-enforcement at the 5% level. As claimed by Klein et al. [41], the specific asset investment in the transaction produces appropriable quasi-rent; it locks the relationship between the two parties and promotes the self-enforcement of the incomplete contract. Farmers who have invested more specific assets in agricultural production expect that the contract can be more effective, which could protect their investment and future gains. This could be achieved by choosing written contracts (results of the select equation of Heckman probit model) or achieved by their spontaneous performance in an informal contract (results of enforcement equation of Heckman). This finding is in line with the research results of Chen and Gao [82], which suggests that enterprises tend to provide specific investments for farmers in the contract relationship, so as to improve the credibility of commitment by enterprises and promote the self-enforcement of the contract.

Regarding the characteristics of farmer householders, a householder being a member of a big clan in the village significantly increases the probability of contract self-enforcement at the 1% level. The age of the householder significantly decreases the probability of contract self-enforcement at the 10% level. Although the education of the householder is shown to be a promoter of self-enforcement, it does not have a significant effect. In our survey area, there are a large number of members of a big clan in the village who share the same family name. Most of them are original residents with a long history of residence, thus accumulating a higher level of social capital. Given the reciprocity between clan members, the farmland transfer is more likely to be informal with customs and favor, in which the relational governance is more conducive to the self-enforcement of contracts [83].

*4.3. Discussion and Limitations*

Combined with the results of informal contract choice and contract self-enforcement, an inference can be drawn that the choice of informal contracts does not necessarily lead to disputes and breaches. It suggests that informal contracts in a rural social environment are a beneficial supplement to the formal system; an enforcement mechanism for informal contracts is also a strong guarantee to reduce the default risk. As for the effect of reputation on contractual arrangements and enforcement, it is not consistent throughout the context of farmland transfer. In other words, the effect of reputation is not reflected as impacting ex-ante contract choices through signaling mechanisms, but mainly as impacting ex-post contract implementation through the penalty effect. Farmers' reputations do not contribute

to informal contract formation by sending sufficient positive signals among them to increase their trust. However, once the informal contract is reached, the reputation mechanism can promote a farmer's self-enforcement by increasing the expected cost of defaults and through invisible supervision within the community. Within a closer social network, the reputation mechanism forms a stronger penalty effect to increase its effectiveness in self-enforcement. By contrast, the social network has a consistent impact on farmers' contract choices and ex-post performance. On the premise of saving transaction costs, a close social network promotes farmers to choose informal contracts through implicit commitments in a favorable relationship. Moreover, it promotes the self-enforcement of informal contracts by establishing relationship governance involving inherent reciprocity, trust, and mutual supervision.

The rules of relational governance in the current acquaintance society in rural China still have strong validity. An informal transfer contract is an important form given the market unpredictability and asymmetric information, which can restrain individual behavior, and it reduces the uncertainty and externalities caused by the unstipulated contents of formal contracts. Based on the validation of the effectiveness of the self-enforcement mechanism, a further important issue is how to enhance the stability of the self-enforcement mechanism. Both personal reputation and social network rely mostly on positive behaviors spontaneously provided by farmers and are subject to factors at the psychological and behavioral levels of farmers, such as their perceptions of inequity, the emphasis on social capital, and adequate consideration of repeated games over time. Therefore, introducing the correct perception of contractual relationships and cultivating farmers' contractual spirit may be the key to enhancing the effectiveness of the self-enforcement mechanism. This is a further issue to be explored. Spontaneous good performance will be positively encouraged by relational rules such as commitment, morality, and mutual benefit. This will help to form a positive habitual preference for individual behavior. Informal rules formed by long-term habitual behaviors are an important driver of institutional change.

Accordingly, it is necessary to reflect on the importance of the consensual nature of the farmland transfer contract. It is not in line with the context of rural China to simply push for unified formal contracts to achieve legitimacy. An important challenge is that the formulation and implementation of the legal system and policy require the support of an informal system and the corresponding spontaneous order; otherwise, it is difficult to form a reasonable, long-term, generally accepted, and proper order [5]. This is especially relevant when it comes to the transaction of farmland. Farmers regard farmland as a personalized form of property and therefore make transactional decisions not only based on economic interests but also on emotional factors. Further research needs to focus on the psychological and behavioral dimensions of farmers under contract relationships. A new analytical paradigm of the contract as a reference point requires close attention to the analysis of farmers' feelings of entitlement and their performance.

There are still several limitations in our study. Primarily, the enlargement of the research area and samples is needed. Our survey is only based on samples from Chengde City, Hebei province due to the time and cost constraints. The single study area inevitably results in the limited generalizability of the findings. Chengde City is an economically underdeveloped region, with predominantly mountainous topography and a late start of farmland transfer. The findings of this study are useful for areas in similar situations, but wider replication requires more careful consideration of the economic development and natural characteristics of the region. In addition, tracking surveys of farmers are needed for further panel studies to explore contract enforcement over a longer period.

## 5. Conclusions and Policy Implications

Based on the data of 402 farmland transfer contracts in Chengde City in the northeastern Hebei province of China, this study examined the choice of informal contracts and their self-enforcement mechanism in farmland transfer. Our findings suggest that there is no ex-ante signaling effect of farmers' reputation on the choice of contract forms. Farmers

choose informal contracts due to the demand for reducing transaction costs and their trust in the other party under the constraints of a close social network. Reputation and social network represent effective ex-post self-enforcement mechanisms in informal contracts. A higher level of reputation promotes self-enforcement by introducing implicit incentives for farmers' social capital accumulation and invisible supervision from the community. A closer social network promotes this by generating relational governance with mutual benefit and trust. The penalty effect of reputational damage is stronger in a closed social network with a higher degree of information symmetry and more efficient dissemination.

With effective informal rules of reputation and social network, informal contracts can play an important role in restraining free-riding and opportunistic behaviors. Self-enforcement of informal contracts can serve as a helpful supplement for formal rules to reduce the default risk. The conclusions may provide references for policies regarding contract arrangements in farmland transfer. Primarily, local governments should pay more attention to creating a supportive environment for the informal rules of farmland transfer. Given the acquaintances-based social system in rural China, "shaking hands instead of contract" can be a more effective strategy to resolve disputes in contracts. Local governments should encourage the establishment of good social norms and the building of personal reputation. A credit supervision and penalty system should be constructed to provide a basis for the implementation of farmland transfer so that farmers can consciously pay attention to reputation accumulation under the expectation of long-term cooperation. Moreover, the rural social network, with an internalized value system of mutual benefit and trust should be maintained to implement supervision and restraints, so as to strengthen the penalty effect of the reputation system. Furthermore, a cooperative governance system incorporating formal and informal rules should be established, including policies, regulations, institutions, and the relational rules of rural society. Policies should standardize the formal regulations of farmland transfer to provide a basis and orientation for the informal rules. This will be an efficient strategy to promote the optimization of contract relationships with affinity, rationality, and general consent.

**Author Contributions:** Conceptualization, H.L. and H.H.; methodology, H.L.; software, H.L. and S.Y.; validation, H.L., H.H. and S.Y.; formal analysis, H.L.; investigation, H.L. and H.H.; data curation, H.L. and S.Y.; writing—original draft preparation, H.L. and S.Y.; writing—review and editing, H.H.; visualization, S.Y.; supervision, H.H.; funding acquisition, H.L. and H.H. All authors have read and agreed to the published version of the manuscript.

**Funding:** This research was funded by: (1) National Natural Science Foundation of China, grant NO. 72103053; (2) Humanities and Social Sciences Research Fund supported by the Ministry of Education of China, grant No.20YJC790059; (3) Natural Science Foundation of Zhejiang Province, China, grant NO. LQ20G030017; (4) National Natural Science Foundation of China, grant NO. 72073119.

**Data Availability Statement:** Not applicable.

**Acknowledgments:** We gratefully acknowledge the financial and administrative support of Hangzhou Normal University and Zhejiang University. The authors also extend great gratitude to the anonymous reviewers and editors for their helpful reviews and critical comments.

**Conflicts of Interest:** The authors declare no conflict of interest.

## Appendix A

**Table A1.** The results of IIA test.

| Categories | Chi2 | df | $p > \text{Chi2}$ | Evidence |
| --- | --- | --- | --- | --- |
| 1 | 4.526 | 20 | 1.000 | For $H_0$ |
| 2 | 18.353 | 21 | 0.627 | For $H_0$ |
| 3 | 1.347 | 20 | 1.000 | For $H_0$ |
| 4 | 4.911 | 19 | 1.000 | For $H_0$ |

**Note:** $H_0$: Odds (outcome-j vs. outcome-k) are independent of other alternatives.

**Table A2.** VIF of variables.

| Variable | VIF | 1/VIF |
|---|---|---|
| Social network | 3.20 | 0.31 |
| Rent prepayment clause | 2.50 | 0.40 |
| Transfer rent | 2.35 | 0.43 |
| Contract duration | 2.12 | 0.47 |
| Transfer modes | 2.11 | 0.47 |
| Education of householder | 1.43 | 0.70 |
| Transferred area | 1.33 | 0.75 |
| Age of householder | 1.28 | 0.78 |
| Rent adjustment clause | 1.24 | 0.81 |
| Farmer's trust | 1.24 | 0.81 |
| Proportion of agricultural income | 1.20 | 0.84 |
| Information access | 1.16 | 0.87 |
| Agricultural durable assets | 1.11 | 0.90 |
| Farmer's reputation | 1.10 | 0.91 |
| Number of transfer contracts | 1.10 | 0.91 |
| Big clan member | 1.07 | 0.94 |
| Mean VIF | 1.60 | - |

**Table A3.** The results of Mprobit model of contract form choice.

| Variables | Mprobit (Based on WU) | | |
|---|---|---|---|
| | **(1) Y = OB** | **(2) Y = OT** | **(3) Y = WD** |
| Farmer's reputation | −0.063 | −0.134 ** | −0.036 |
| | (0.058) | (0.064) | (0.035) |
| Social network | −1.587 *** | −1.631 *** | −0.339 * |
| | (0.283) | (0.288) | (0.177) |
| Transferred area | −0.058 | −0.049 | 0.008 |
| | (0.038) | (0.034) | (0.015) |
| Transfer rent | −0.001 | 0.001 | 0.004 *** |
| | (0.001) | (0.000) | (0.000) |
| Transfer modes | −2.160 *** | −1.547 *** | −1.832 *** |
| | (0.527) | (0.497) | (0.408) |
| Agricultural durable assets | −0.024 * | −0.033 ** | −0.011 |
| | (0.013) | (0.013) | (0.014) |
| Proportion of agricultural income | 1.339 | 0.911 | 1.100 |
| | (0.817) | (0.779) | (0.747) |
| Information access | −0.000 | −0.000 | 0.000 |
| | (0.000) | (0.000) | (0.000) |
| Number of transfer contracts | −0.037 | −0.039 | −0.110 |
| | (0.234) | (0.198) | (0.323) |
| Farmer's trust | −0.652 | 0.149 | 1.414 *** |
| | (0.456) | (0.448) | (0.336) |
| Big clan member | −0.378 | −0.835 ** | −0.122 |
| | (0.413) | (0.408) | (0.309) |
| Age of householder | 0.023 | −0.056 ** | −0.037 ** |
| | (0.029) | (0.024) | (0.015) |
| Education of householder | −0.082 | −0.226 *** | −0.109 * |
| | (0.097) | (0.086) | (0.064) |
| Constant | 4.576 ** | 8.955 *** | 0.091 |
| | (1.992) | (1.780) | (1.245) |
| Prob > chi2 | | 0.000 | |
| Observation | | 403 | |

Notes: ***, **, * represent significance at levels of 1%, 5%, and 10%, respectively.

## Notes

[1]  Source: Chengde Municipal government (http://www.chengde.gov.cn/cdgk/2007-10/12/content_2024.html, accessed on 1 December 2015) and Agricultural Economic Statistics Report of Chengde Municipal Bureau of Statistics in 2014. However, there is no detailed information of the number of contracts in the period of 2010–2014, and oral contracts are not included in the statistics.

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
