# Peer review of "Reputation Effect on Contract Choice and Self-Enforcement: A Case Study of Farmland Transfer in China"

_land, doi:10.3390/land11081296_

Round 1
Reviewer 1 Report
This is an interesting attempt to estimate the effect of reputation on contract choice (formal vs. informal) and self-enforcement in land-transfer contracts based on a small sample of 402 contracts in one administrative subdivision (Chengde City) in one Chinese province (Hebei in the Northeast). Given the small sample size (both in absolute terms and relative to the total number of contracts in Hebei – see line 278), it is necessary to discuss the representativeness (or lack thereof) of the sample and the findings.
I detect a disturbing carelessness in model specifications. Thus, models (3) and (4) are corrupted (they contain wrong symbols); under model (5) the variable Inforcement does not exist: this should probably be Informal; model (6) contains an unexplained symbol ∅ (the usual meaning of this symbol is the empty set, but this is obviously not what model (6) needs).
Signaling is an important concept in your analysis. Please ensure careful and thorough explanation of this concept in the conceptual part. Even more important, explain how the signaling effect is reflected in your regression tables. What variables and numbers in the analytical tables represent the signaling effect? How does it behave as a function of other variables.
The variable “reputation” is continuous (the number of times the respondent was called upon to mediate disputes in the village). So, reputation can increase, decrease, or remain constant. You use two very curious expressions “reputation devaluation” and then, in a couple of places, even more curiously “reputation depreciation”. These pseudo-terms are meaningless in English. Please change them so that the reader can clearly understand what you mean.
Section 3.2.2 “Other control variables” should be better organized. Somewhere in this section say that the variable definitions are given in Table 2. The long first paragraph should be broken into several paragraphs, each for a different category of variables. It will help the reader if the variables in each category were presented in bulleted lists, each bullet followed by the appropriate definition. In addition, the terms “transferred scale, transferred rent, and transferred way” are syntactically confusing. Particularly bad is “transferred way”: I suggest you use “transfer modes” or “transfer methods” (“transfer”, not “transferred”). For “transferred scale” it may be better to use “transferred area”. For “transferred rent” consistently use “contract rent” (without jumping from one term to the other). For “transferred times” use “number of transfer contracts”.
Why is “agricultural durable assets” included in two categories (household characteristics and dependence on farming)? How does this affect your regression analyses?
Line 373: “In addition to the common explanatory variables” – what makes them “common” and how are they different from the others? The above comments to list organization should be applied also to these “additional” variables.
In Table 3 of descriptive statistics, you have to indicate which differences between oral and written contracts are statistically significant. In the body of the text, whenever you compare estimates for the two type of contracts, always indicate whether the difference is statistically significant or not. If the difference is not statistically significant, it makes no sense to talk about “greater” or “smaller”, “higher” or “lower”.
In Figure 2, clearly differentiate between districts covered by the survey and the other districts in the City. Perhaps non-participating districts should be shown without color.
In all numbers in the article, use comma separators between hundreds, thousands, etc.
Line 281: “more than 800 disputes and defaults in Chengde City” – more than 800 out of how many total transfer contracts in Chengde City?
Line 44: “restore the market order of farmland transfer” – in what way has the market order been broken?
Line 99: Liu and Lv (2017) does not exist in the list of references.
On lines 407-408, the unit of measurement is yuan/mu, not yuan.
On line 474, instead of “The test of IIA”, write “The test IIA” or “The IIA test”. Here and in the title of Table 4 provide a reference to this test, and in the body of the text (or a footnote) perhaps briefly say what this test does and why it is needed.
Lines 328-329: omit the parenthetical clause with examples – it is unnecessary and confusing.
Line 338-339: “The tightness of social network decreases along with increasing kinship and geographic distance” – this sounds very odd. What exactly to you mean by “increasing kinship”? Kinship between subject and brother is higher than kinship between subject and neighbor? If so, then network tightness INCREASES with increasing kinship. Please check and amend as necessary.
On line 729 say again where exactly in China Chengde City is. Also, remember to capitalize “City” everywhere.
On line 769 you mention the “autonomy principle of land transfer”. What is this exactly?
On lines 741-742, you write: “The key findings show that informal contract of farmland transfer has advantages of low transaction cost and high flexibility.” The statement is, of course, correct but which of you findings show that this is indeed so?
Line 420: Specify that the “dummy” variable is binary or multinomial.
Problems with English:
Line 21 (abstract): “which fits with the status of farmland transfer embedded in rural acquaintance society” – unclear sentence
Line 322: “Refers to Lei and Li (2012)” – “Referring to Lei and Li (2012)”
Line 333: “by differentiate” (????)
Line 335: “ordered categories” instead of “orderly categories”
Line 341: “Circled networks” – do you mean “circular networks”?
Line 431: “the most completed contract form of WU” change to “most frequently completed contact form WU” (note: WU, not of WU).
Line 468: change “implicates” to “implies”.
Line 732: “relational governance with mutually beneficial and trust” – change to “mutual benefit and trust” (as on line 765).
Line 768: endue (?)
End of review comments
Reviewer 2 Report
About the submission with the title "Reputation Effect on Contract Choice and Self-enforcement: A Case Study of Farmland Transfer in China" I have the following comments:
The abstract should be rewritten with the main motivations, gaps in the literature that justify the research, objectives, methodologies, novelties and main insights. Another question to be improved is the presentation of some concepts for the readers. For example, what is "self-enforcing mechanisms"? Or, what means "The results show that a good reputation encourages contract self-enforcement by the penalty effect of increasing the anticipated default cost."?
In section 3, it could be important to present the characteristics of the sample, to better understand the robustness of the results presented in the several tables.
I have some doubts about the approach presented in the section 3.3.1. Maybe a cluster and factor analyses could be more adjusted, namely to reduce the number of variables.
In fact, the consideration of many variables in the regressions of tables 5 and 7 may bring problems of multicollinearity. On the other hand, it would be expected in these regression concerns (statistical tests) with also the heteroscedasticity, autocorrelation (including spatial), endogeneity, normality, linearity, ....
The potential problems may compromise the robustness of the results.
Round 2
Reviewer 2 Report
The authors improved the paper.